# Adaptation of metabolite leakiness leads to symbiotic chemical exchange and to a resilient microbial ecosystem

**Jumpei F. Yamagishi**[1], **Nen Saito**[2,3]*, **Kunihiko Kaneko**[1,3]*

**1** Graduate School of Arts and Sciences, The University of Tokyo, Meguro-ku, Tokyo, Japan, **2** Exploratory Research Center on Life and Living Systems, National Institutes of Natural Sciences, Okazaki, Aichi, Japan, **3** Research Center for Complex Systems Biology, Universal Biology Institute, The University of Tokyo, Meguro-ku, Tokyo, Japan

* saito@ubi.s.u-tokyo.ac.jp (NS); kaneko@complex.c.u-tokyo.ac.jp (KK)

**Data Availability Statement:** Computer codes used in this study are available at the following URL: https://github.com/JFYamagishi/microbial_ potlatch.

## Abstract

Microbial communities display remarkable diversity, facilitated by the secretion of chemicals that can create new niches. However, it is unclear why cells often secrete even essential metabolites after evolution. Based on theoretical results indicating that cells can enhance their own growth rate by leaking even essential metabolites, we show that such "leaker" cells can establish an asymmetric form of mutualism with "consumer" cells that consume the leaked chemicals: the consumer cells benefit from the uptake of the secreted metabolites, while the leaker cells also benefit from such consumption, as it reduces the metabolite accumulation in the environment and thereby enables further secretion, resulting in frequency-dependent coexistence of multiple microbial species. As supported by extensive simulations, such symbiotic relationships generally evolve when each species has a complex reaction network and adapts its leakiness to optimize its own growth rate under crowded conditions and nutrient limitations. Accordingly, symbiotic ecosystems with diverse cell species that leak and exchange many metabolites with each other are shaped by cell-level adaptation of leakiness of metabolites. Moreover, the resultant ecosystems with entangled metabolite exchange are resilient against structural and environmental perturbations. Thus, we present a theory for the origin of resilient ecosystems with diverse microbes mediated by secretion and exchange of essential chemicals.

## Author summary

How can diverse species or strains coexist in microbial communities? Besides the fittest strain under isolation conditions, many others coexist. Population dynamics with appropriate cell-to-cell interaction would provide such diversity: for example, secretion of beneficial metabolites from the fittest strain could feed others and enable the coexistence. But how this interaction is achieved and maintained remains elusive. Following a recent theoretical study suggesting that appropriate leakage of essential metabolites is beneficial to the leaking cells themselves, we demonstrate that mutualistic relationships among diverse

**Funding:** KK was partially supported by a Grant-in-Aid for Scientific Research (A) (20H00123) and Grant-in-Aid for Scientific Research on Innovative Areas (17H06386) from the Ministry of Education, Culture, Sports, Science and Technology (MEXT) of Japan. he funders had no role in study design, data collection and analysis, decision to publish, or preparation of the manuscript.

**Competing interests:** The authors have declared that no competing interests exist.

species can be established as a result of cell-level adaptation of metabolite leakiness: each species cross-feeds others by secreting out essential metabolites for its own sake, which are usefully consumed by others, as if cells exchange gift chemicals with each other. In this case, exchange of metabolites is entangled, which leads to the coexistence of diverse microbes, and the resultant ecosystems are resilient against external perturbations including removal of each coexisting species. A new basis for diverse, complex microbial ecosystems is thus presented.

## Introduction

In microbial communities, diverse species or strains in bacteria, archaea, and fungi coexist [1–4]; while those microbes secrete and exchange hundreds of (essential) metabolites such as amino acids [5, 6], sugars [5, 7], organic acids [8, 9], vitamins [10, 11], nucleotides [5, 12], and intermediates of the TCA cycle [13, 14]; an archaeon transfers lipids and possibly even ATP [15, 16]. Metabolite secretion and exchange can allow for the coexistence of various microorganisms rather than the dominance of a single fittest species that excludes all others [4, 17, 18]. Even when only a single resource is available, metabolite secretion can create new niches, and thereby, more diverse species can coexist [17, 19, 20]. Moreover, the metabolite exchange is often reported to shape mutually symbiotic relationships, rather than parasitism, among the coexisting microbial species [7, 21–24].

Nonetheless, the evolutionary origin(s) of secretion and exchange of essential or costly metabolites that support microbial coexistence and symbiosis remain enigmatic. For example, a constructive laboratory experiment revealed that stronger cells (i.e., cells with higher glutamine synthetase activity) coexist with weaker cells, via leakage of glutamine synthesized by the former [25]. Lenski *et al.* stressed the importance of chemical leakage by proposing the black queen hypothesis [26, 27]: if one species leaks some resource chemical, other species can take advantage of such leakage and will become dependent on it through gene loss, thereby leading to the evolution of metabolic dependency. However, these previous studies did not examine whether leakage is beneficial for leaker cells themselves. Rather, leakage is simply assumed to be inevitable because of the permeability of their cell membranes [27–29], albeit disadvantageous it may be. If this is the case, there is no reciprocity between the leaker cells and the other cells; then, why have the leaker cells not evolved to decrease the leakiness and not dominated the ecosystem?

The maintenance of metabolite-mediated symbiosis is also a theoretical concern. Since the leaked metabolites are "public goods" that are freely available to all coexisting species, such symbiosis seems to be vulnerable to the invasion of cheaters. Nevertheless, cooperation via leaked metabolites is promoted in nature [30] and in laboratory evolution [31].

We recently proposed through a theoretical model that leakage of even essential metabolites can promote the growth of the leaker cells in isolated conditions [32]. The balance between chemical synthesis and growth-induced dilution with an autocatalytic reaction process in a cell generally leads to this phenomenon termed *leak advantage*. In reality, microbes leak a variety of essential metabolites including central metabolic intermediates, even in isolated conditions [14]. This leak advantage can explain why leakage of even essential metabolites is preserved or acquired through evolution [30] and may provide a new perspective on metabolite-mediated microbial ecology.

In the present paper, we examined whether and how cell-cell interactions mediated by secreted metabolites can lead to stable coexistence of diverse microbial species (or strains or

mutants), through numerical simulations of models that include the dynamics of intracellular states and environmental states. We first show that "leaker" and "consumer" species (i.e., cells that benefit by leaking some chemicals and those that benefit by consuming them, respectively) can immediately develop a mutualistic relationship. In this case, the leaker species leak metabolites for their own sake; hence, cheaters that exploit altruistic behavior cannot emerge. We then explore the conditions under which such "leaker-consumer mutualism" is observed and stable.

Based on the idea of leaker-consumer mutualism, we will further show that when each coexisting cell species optimizes its own growth (which may result from adaptation within a generation or evolution over generations), the coexistence of diverse species is achieved, and the overall growth rate of the microbial community is enhanced. This novel scenario for symbiosis among diverse species will explain why the single "fittest" species does not dominate as a result of evolution. Furthermore, we will show that systems with exchange of metabolites among diverse species are resilient against external perturbations. Finally, we discuss the possible relevance of the present results for experimental characterization of the resilience of microbial ecosystems.

## Model

Let us consider a situation where cells that contain $n$ kinds of chemical components (metabolites and enzymes) coexist in a common environment. Each cellular state is represented by the set of concentrations of these chemicals, as in previous studies [33–35]. Intracellular chemical reactions convert the externally-supplied nutrient into biomass for cell growth, while the cellular volume growth dilutes intracellular chemical concentrations. Those cells can exchange metabolites with the other cells via a well-mixed (spatially unstructured) environment (see Fig 1 and Table 1).

The state of cell $\alpha$ is expressed by the concentrations of the $n$ components, $\mathbf{x}^{(\alpha)} = {}^{t}(x_0^{(\alpha)}, x_1^{(\alpha)}, \cdots, x_{n-1}^{(\alpha)})$. In each cell, chemical $i$ is synthesized and decomposed by a set of intracellular reactions at the rate $F_i^{(\alpha)}(\mathbf{x}^{(\alpha)})$ and is exchanged with the environment at the rate $f_i^{(\alpha)}(\mathbf{x}^{(\alpha)}; D_i^{(\alpha)}, x_i^{(\text{env})})$, where $x_i^{(\text{env})}$ represents the concentration of chemical $i$ in the environment. Here, out of $n$ chemical components in each network, $n_{\text{enzyme}}$ chemicals are "enzymes" which could be a catalyst or product of each reaction, while the rest of the chemicals, termed as "metabolites," could be diffusible (if $D_i^{(\alpha)}$ is positive) substrates or products of each reaction.

The temporal evolution of the concentration of chemical $i$ in cell $\alpha$, $x_i^{(\alpha)}$, is generally written as

$$\dot{x}_i^{(\alpha)} = F_i^{(\alpha)}(\mathbf{x}^{(\alpha)}) + f_i^{(\alpha)}(\mathbf{x}^{(\alpha)}; D_i^{(\alpha)}, x_i^{(\text{env})}) - \mu^{(\alpha)}(\mathbf{x}^{(\alpha)})x_i^{(\alpha)}$$

where $\mu^{(\alpha)}(\mathbf{x}^{(\alpha)})$ is the volume growth rate of the $\alpha$th cell, and the third term represents the dilution of each chemical owing to the increase in cellular volume. We here discuss the case of passive diffusion, where the flow rate of chemical $i$ is given by $f_i^{(\alpha)}(\mathbf{x}^{(\alpha)}; D_i^{(\alpha)}, x_i^{(\text{env})}) = D_i^{(\alpha)}(x_i^{(\text{env})} - x_i^{(\alpha)})$. $D_i^{(\alpha)}$ is a non-negative parameter characterizing the flow rate of each metabolite $i$, which we call the diffusion coefficient: it can be interpreted as the permeability of the cell membrane to each metabolite, or the "coarse-grained" abundance of transporter/channel/porin that is controlled by gene regulation (i.e., $D_i^{(\alpha)}$ is proportional to their abundance, which is determined genetically). If $f_i^{(\alpha)}$ is positive, then chemical $i$ flows into cell $\alpha$ from the environment, and if it is negative, $i$ is leaked out. Note that we define *leak-advantage chemicals* for species $\alpha$ such that an (infinitesimal) increase in their leakage promotes the growth of species $\alpha$ [32].

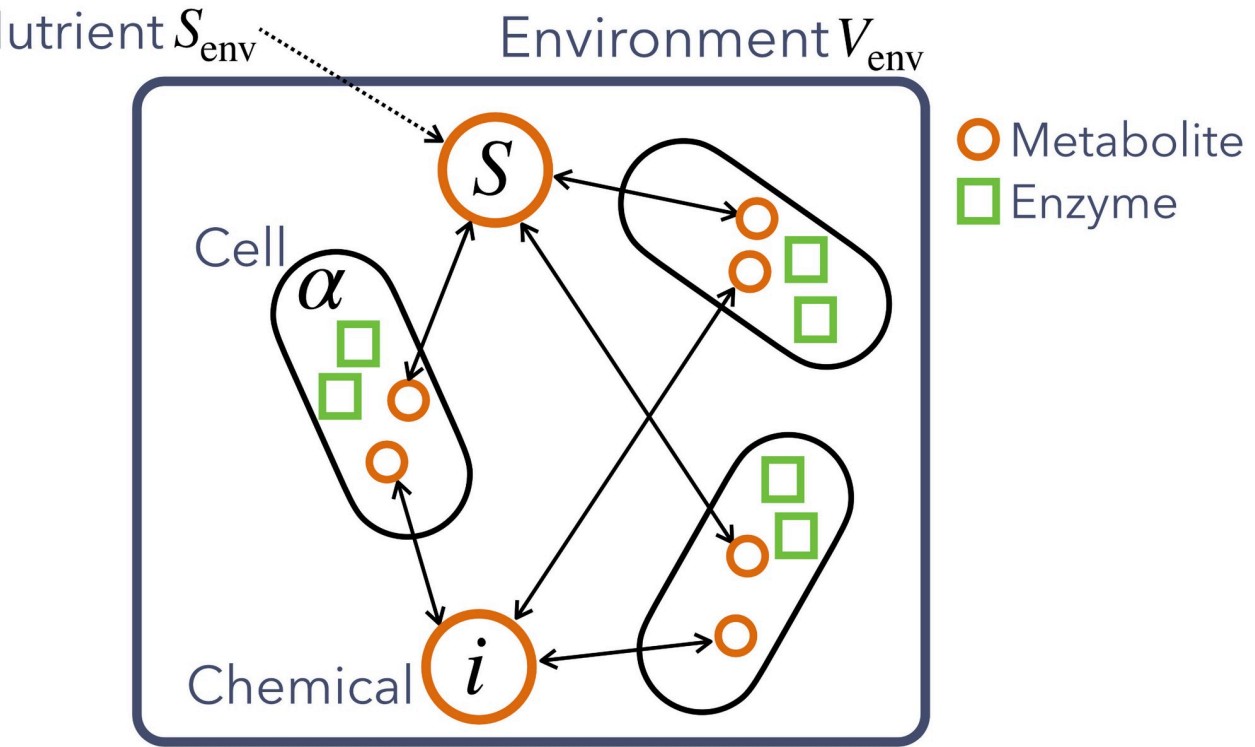

**Fig 1. Schematic illustration of the model for microbial community with metabolite exchange.** Each cell species $\alpha$ has a chemical reaction network that transforms a single nutrient $S$ transported from the environment for cell growth. The nutrient $S$ is supplied to the environment from the exterior at the rate $D_S^{(\text{env})}(S_{\text{env}} - x_S^{(\text{env})})$. Among $n$ chemicals, metabolites (orange circles) are diffusible and exchanged by coexisting species via the environment, while enzymes (green squares) are not.

**Table 1. Parameters and variables in our model.**

| Symbol | Description |
|---|---|
| Parameters | |
| $N$ | Number of cell species |
| $n$ | Number of chemicals (metabolites and enzymes) |
| $n_{\text{enzyme}}$ | Number of enzymes |
| $S_{\text{env}}$ | Abundance of externally supplied nutrient |
| $V_{\text{env}}$ | Volume ratio of the environment to the whole cell population |
| $R_{\text{deg}}$ | Rate of chemical degradation in the environment |
| $D_S^{(\text{env})}$ | Diffusion coefficient of nutrient between the environment and its exterior |
| $D_S$ | Diffusion coefficient for uptake of nutrient chemical by cell species |
| $\rho$ | Average number of reactions per chemical |
| Variables | |
| $x_i^{(\alpha)}$ | Concentration of chemical $i$ in cell species $\alpha$ |
| $x_i^{(\text{env})}$ | Concentration of chemical $i$ in the environment |
| $\mu^{(\alpha)}$ | Growth rate of cell species $\alpha$ |
| $D_i^{(\alpha)}$ | Diffusion coefficient of non-nutrient metabolite $i$ for cell species $\alpha$ |

Note that the dimension of the "diffusion coefficients" $D_i$ is [time$^{-1}$], which is different from that of the diffusion coefficient in the ordinary sense, [space$^2 \times$ time$^{-1}$]. One can interpret $D_i$ as characterizing the membrane permeability to each metabolite $i$ or the abundance of transporter proteins for $i$. In our model, $D_i$ evolves by cell-level adaptation: in the situations without such adaptation, $D_i$ is fixed.

To account for cell-cell interactions due to the transport of chemicals through the environment, the time evolution of the external concentrations $\mathbf{x}^{(\text{env})}$ is given as

$$
\begin{aligned}
\dot{x}_i^{(\text{env})} &= -\sum_\alpha p_\alpha f_i^{(\alpha)}(\mathbf{x}^{(\alpha)}; D_i^{(\alpha)}, x_i^{(\text{env})})/V_{\text{env}} - R_{\text{deg}} x_i^{(\text{env})} \\
&= \sum_\alpha p_\alpha D_i^{(\alpha)}(x_i^{(\alpha)} - x_i^{(\text{env})})/V_{\text{env}} - R_{\text{deg}} x_i^{(\text{env})},
\end{aligned}
\tag{1}
$$

when chemical $i$ is not a nutrient. If chemical $i$ is a nutrient, it is supplied to the environment via simple diffusion, so that the term $D_i^{(\text{env})}(S_{\text{env}} - x_i^{(\text{env})})$ is added to the right-hand side of the above Eq (1). In the external medium, the secreted components slowly degrade or flow out of the medium at the rate $R_{\text{deg}} x_i^{(\text{env})}$. The volume of the environment relative to the total volume of all the coexisting cells is designated as $V_{\text{env}}$.

Now, we consider the temporal change of the population fraction of cell species $\alpha$, given by $p_\alpha$. If the intracellular dynamics are faster than the population dynamics, we can assume that $\mathbf{x}^{(\alpha)}$ reaches the stationary state, from which $\mu^{(\alpha)}$ is determined. Then, the growth of the population fraction is given by $\mu^{(\alpha)} p_\alpha$; noting that the fractions of all species satisfy $\Sigma_\alpha p_\alpha = 1$ holds, the population dynamics are given by the replicator-type equation

$$
\dot{p}_\alpha = (\mu^{(\alpha)} - \bar{\mu})p_\alpha
\tag{2}
$$

where $\bar{\mu} \equiv \sum_\alpha p_\alpha \mu^{(\alpha)}$ is the average growth rate of all species [36].

We then investigated the steady state of the population dynamics of multiple species with different reaction networks, and examined whether they can coexist in a common environment.

## Results

### A simple example of leaker-consumer mutualism: Symbiosis between two species

In order to exemplify leaker-consumer mutualism, we first consider the simplest situation: symbiosis between two cell species in which the leaker cells secrete an essential metabolite and the consumer cells consume it to facilitate their growth.

As one of the simplest examples, the network structure of the example in [32]—a simple reaction network that consists of substrate $S$, enzyme $E$, ribosome rb, metabolites $M_1$ and $M_2$, biomass BM—is adopted both for the leaker and consumer cells in this subsection (Fig 2A), whereas metabolic networks simplified from real data can also show the leak advantage (see S1 Fig). The equations for the reactions are given below, and the rate constants given below are different for the leaker and consumer cells:

$$
\begin{aligned}
\dot{x}_S^{(\alpha)} &= -k_{S \to M_1}^{(\alpha)} x_S^{(\alpha)} x_E^{(\alpha)} - k_{S \to M_2}^{(\alpha)} x_S^{(\alpha)} + D_S\left(S_{\text{env}} - x_S^{(\alpha)}\right) - \mu^{(\alpha)} x_S^{(\alpha)}, \\
\dot{x}_{M_1}^{(\alpha)} &= k_{S \to M_1}^{(\alpha)} x_S^{(\alpha)} x_E^{(\alpha)} - \left(k_{M_1 \to \text{rb}}^{(\alpha)} + k_{M_1 \to E}^{(\alpha)}\right) x_{M_1}^{(\alpha)} x_{\text{rb}}^{(\alpha)} + D_{M_1}\left(x_{M_1}^{(\text{env})} - x_{M_1}^{(\alpha)}\right) - \mu^{(\alpha)} x_{M_1}^{(\alpha)}, \\
\dot{x}_{\text{rb}}^{(\alpha)} &= k_{M_1 \to \text{rb}}^{(\alpha)} x_{M_1}^{(\alpha)} x_{\text{rb}}^{(\alpha)} - \mu^{(\alpha)} x_{\text{rb}}^{(\alpha)}, \\
\dot{x}_E^{(\alpha)} &= k_{M_1 \to E}^{(\alpha)} x_{M_1}^{(\alpha)} x_{\text{rb}}^{(\alpha)} - \mu^{(\alpha)} x_E^{(\alpha)}, \\
\dot{x}_{M_2}^{(\alpha)} &= k_{S \to M_2}^{(\alpha)} x_S^{(\alpha)} - k_{M_2 \to \text{BM}}^{(\alpha)} x_{M_2}^{(\alpha)} x_E^{(\alpha)} - \mu^{(\alpha)} x_{M_2}^{(\alpha)},
\end{aligned}
$$

where the growth rate is defined as the rate of synthesis of biomass BM from its precursor $M_2$, such that $\mu^{(\alpha)}(\mathbf{x}^{(\alpha)}) \equiv k_{M_2 \to \text{BM}}^{(\alpha)} x_{M_2}^{(\alpha)} x_E^{(\alpha)}$. Although all chemicals—$S$, $E$, rb, $M_1$, and $M_2$—are

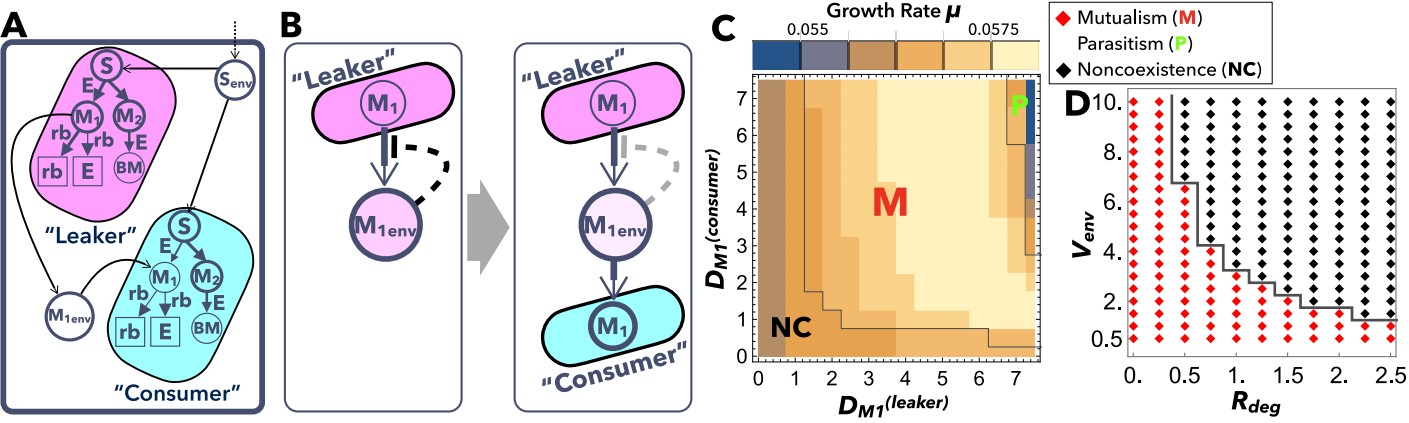

**Fig 2. Example of leaker-consumer mutualism between two cell species.** (A) A simple example of the mutualism between the leaker (left) and consumer (right) cells. Both have the same network structure that consists of substrate $S$, enzyme $E$, ribosome rb, metabolites $M_1$ and $M_2$, and biomass BM, with different rate constants. (B) Schematic illustration of leaker-consumer mutualism. When only a leaker cell is present, the secreted chemical accumulates in the environment and inhibits further secretion (left). The coexistence of other consumer cells is beneficial for both cells as it reduces the concentration of the leaked chemical in the environment (right). (C) Phase diagram of symbiosis depending on $D_{M_1}^{(\text{leaker})}$ and $D_{M_1}^{(\text{consumer})}$. Regions M (red), P (green), and NC (black) are delineated by gray lines and represent mutualism, parasitism, and noncoexistence, respectively. The environment volume ratio $V_{\text{env}}$ and the degradation rate in the environment $R_{\text{deg}}$ are both set at unity. The color denotes the growth rate $\mu$, where a brighter color corresponds to a higher $\mu$. $\mu$ with $D_{M_1}^{(\text{consumer})} = 0$ is just the growth rate of the leaker cells in isolated conditions, $\mu_{\text{iso}}^{(\text{leaker})}$; in region M, it is smaller than $\mu$ at the corresponding $D_{M_1}^{(\text{leaker})}$ value in the panel. (D) Phase diagram of symbiosis depending on the environmental parameters, $R_{\text{deg}}$ and $V_{\text{env}}$. The diffusion coefficients of $M_1$ are fixed as $D_{M_1}^{(\text{leaker})} = D_{M_1}^{(\text{consumer})} = 4$. Red and black diamonds are delineated by gray lines and represent mutualism and noncoexistence, respectively. The rate constants are set as: $k_{S \to M_1}^{(\text{leaker})} = 1$, $k_{S \to M_1}^{(\text{consumer})} = 0.4$, $k_{M_1 \to \text{rb}}^{(\text{leaker})} = k_{M_1 \to E}^{(\text{leaker})} = k_{S \to M_2}^{(\text{leaker})} = 1$, $k_{M_1 \to \text{rb}}^{(\text{consumer})} = k_{M_1 \to E}^{(\text{consumer})} = k_{S \to M_2}^{(\text{consumer})} = 2$, $k_{M_2 \to \text{BM}}^{(\text{leaker})} = k_{M_2 \to \text{BM}}^{(\text{consumer})} = 0.01$, so that the leaker's growth rate in isolation $\mu_{\text{iso}}^{(\text{leaker})}$ with optimal diffusion coefficient $D_{M_1}^{(\text{leaker})}$ is higher than the consumer's growth rate in isolation $\mu_{\text{iso}}^{(\text{consumer})}$ with $D_{M_1}^{(\text{consumer})} = 0$. The other parameters are set as $S_{\text{env}} = 1.0, D_S^{(\text{env})} = 10.0, D_S = 1.0$.

necessary for the growth of cells with this reaction network, certain levels of leakage of metabolite $M_1$ can promote the growth of cell $\alpha$ with a relatively large rate constant $k_{S \to M_1}^{(\alpha)}$. This is because $E$, rb, and $M_1$ constitute an autocatalytic module, which may work excessively. This is a mechanism for leak advantage, "flux control mechanism" proposed in [32]. It should be noted that even if the chemical reactions are reversible instead of irreversible ones assumed here, the leak advantage can generally appear (see S2 Fig). In this example of symbiosis between two cells, the rate constants of the leaker and consumer cells are set at $k_{S \to M_1}^{(\text{leaker})} = 1.0 > k_{S \to M_1}^{(\text{consumer})} = 0.4$; thus, the leakage (uptake) of $M_1$ is beneficial only for the former (latter) (Fig 2A).

Our numerical simulations show that the mutualism between the leaker and the consumer cells is achievable under certain conditions, as schematically illustrated in Fig 2B. When only the leaker cells that gain a leak advantage by secreting a chemical are present in the system, the secreted chemical accumulates in the environment so that further secretion becomes difficult due to the loss of the concentration gradient (see the left panel in Fig 2B). Since the leaked chemical is an essential metabolite and not a waste product, other species can use it for their own growth in most cases. This consumption of the leaked chemical by other cell species is also beneficial for the leaker cells, as it reduces the accumulation of the leaked chemicals in the medium (see the right panel in Fig 2B).

Hence, the leaker and consumer cells coexist through this leaker-consumer mutualism, when diffusion coefficients range within certain values indicated as Region M (Mutualism) in Fig 2C. Under such conditions, the growth rates of the two cells are equal, and are higher than that of either cell species grown in isolation (see $D_{M_1}^{(\text{consumer})} = 0$ in Fig 2C which represents the growth rates of the leaker cells in isolation). However, when $D_{M_1}^{(\text{leaker})}$ and/or $D_{M_1}^{(\text{consumer})}$ are

small, the growth rate of the consumer cells cannot increase to that of the leaker cells, and thus, only the leaker cells exist in the environment (Noncoexistence; Region NC in Fig 2C). In contrast, when $D_{M_1}^{(\text{leaker})}$ and $D_{M_1}^{(\text{consumer})}$ are large, the leaker and consumer cells can still coexist, but their growth rate is lower than that observed when leaker cells grow in isolation. Thus, parasitism, rather than mutualism, is realized in such cases (Region P in Fig 2C). This is because excess leakage of a necessary chemical $M_1$ is disadvantageous to the leaker. Fig 2C, however, also indicates that the fastest growth is achieved by mutualism between the two types of cells. Accordingly, if both cells adaptively alter their diffusion coefficients, parasitic coexistence is excluded, and mutualistic coexistence is expected to emerge. Notably, such cell/individual-level growth optimization via adaptive changes of diffusion coefficients of each cell spontaneously leads to optimal growth at the community/ecosystem level.

Additionally, Fig 2D reveals that the leaker-consumer mutualism is achieved if $R_{\text{deg}}$ and $V_{\text{env}}$ are not very large, that is, if the secreted chemical is efficiently transported to the other cell. However, when the degradation rate $R_{\text{deg}}$ or environment size $V_{\text{env}}$ is too large to allow for sufficient accumulation of the secreted metabolite in the environment, the cells no longer coexist, and only the leaker cell survives (see also [25]).

## Symbiosis among randomly generated networks because of cell-level adaptation

To investigate the possibility of leaker-consumer mutualism and symbiosis among more cell species with diverse chemicals, we further considered a model that includes various cell species with randomly generated catalytic networks consisting of metabolites and enzymes (see Fig 3A and S3 Fig). The transport of chemicals from one cell species to another can be bidirectional if their membranes are permeable to diverse chemicals, which may lead to a complex symbiotic relationship. For simplicity, we considered reaction networks including only two-body catalytic reactions such as $i + k \rightarrow j + k$ with a catalyst $k$ and equal rate constants (set at unity; then, the reaction rate is given by $x_i x_k$), and only a single nutrient (chemical 0) is supplied externally.

We first generated a "species pool" containing $N = 50$ randomly generated networks with no specific structure: each network consists of $\rho n$ randomly chosen catalytic reactions where an enzyme catalyzes conversion from a metabolite to another metabolite or enzyme. Note that a sufficient fraction of such random networks as in Fig 3A and S3 Fig show leak advantage, even though they are not designed to involve specific structures as in Fig 2 (see also [32]).

Then, these $N$ species were added to the environment one by one. At the time of the addition, the invading species optimizes the diffusion coefficients of non-nutrient metabolites to maximize its growth rate under the environmental state $\mathbf{x}^{(\text{env})}$, where other cell species exist before the invasion. After the addition of the new species, the population dynamics of Eq (2) are computed over a sufficiently long period $T$, until the population distribution reaches a steady state (species with population ratio smaller than the threshold value $p_{\text{min}} = 2.5 \times 10^{-4}$ will be eliminated). After this procedure, each surviving species gradually and simultaneously alters its diffusion coefficients for non-nutrient metabolites over period $T$ so that its own growth rate increases. This process of invasion and cell-level adaptation is repeated for all $N$ species (see S1 Text for details).

To examine whether symbiosis among cells with randomly chosen networks can be achieved as a consequence of cell-level adaptation of leakage and uptake, the above model was numerically studied. As shown in Fig 3B, invasions can occasionally reduce the number of cell species in the environment, but generally, the number and growth rates of coexisting species increase. Consequently, multiple cell species could steadily coexist by exchanging multiple

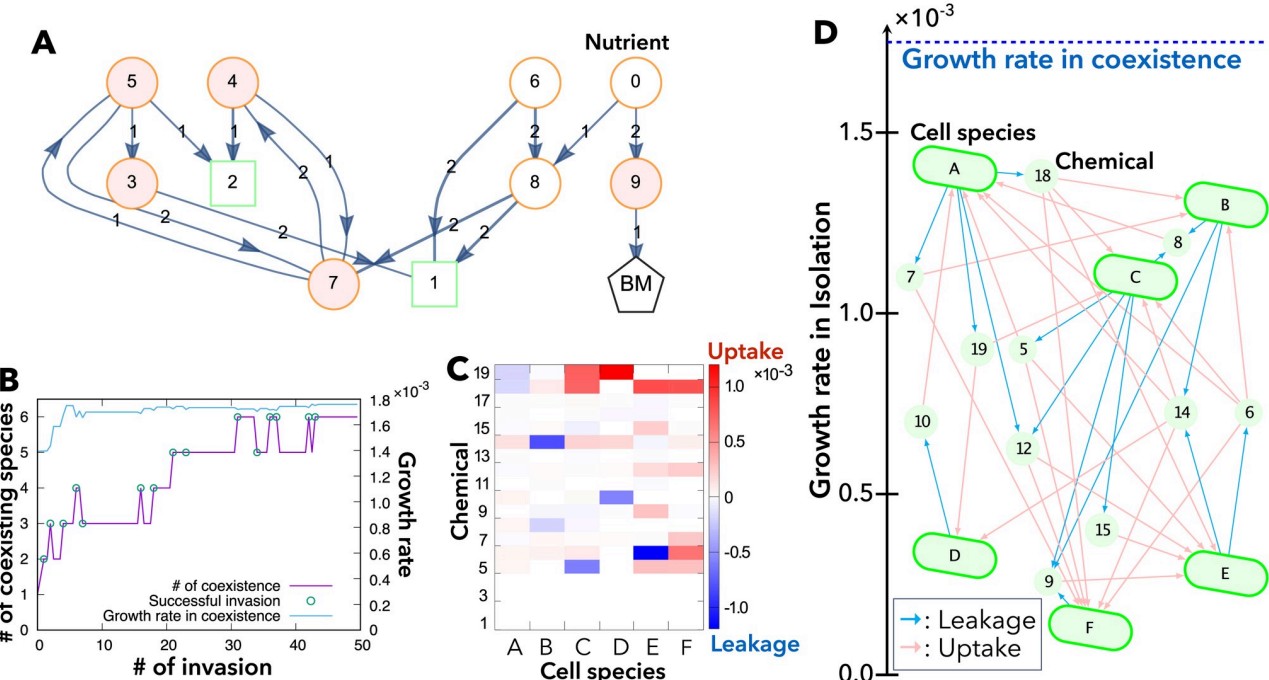

**Fig 3. Example of symbiosis with metabolic exchange via the environment.** (A) An example of randomly generated networks with $n = 10$. The enzyme labeled on each arrow catalyzes the conversion of the metabolite at the arrowtail to the metabolite or enzyme at the arrowhead. Among $n$ chemicals, chemicals 1 and $n_{enzyme} = 2$ are enzymes (green squares) and the nutrient chemical 0 and chemicals $n_{enzyme} + 1 = 3$ to $n - 1 = 9$ are metabolites (orange circles). The leak-advantage metabolites in isolation conditions (chemicals 3–5, 7, and 9) are highlighted by pink. See also S3(B) Fig for the network of cell species B in panels (C) and (D). (B) Time series of the number of coexisting species through successful invasions by new species and the growth rate of coexisting cell species. (C) Plot of leakage (blue) and uptake (red) fluxes of non-nutrient chemicals from each cell species A-F. (D) Structure of metabolic exchange among six cell species that have different growth rates in isolation. The vertical axis represents the growth rate of each cell species $\alpha$ in isolation, $\mu_{iso}^{(\alpha)}$. Cyan and pink arrows indicate the leakage and uptake of each chemical component, respectively. Symbiosis among multiple species increases the growth rate to $\mu_{symbiosis}$ (as indicated on the top), which is higher than the growth rate of each cell species in isolation, $\mu_{iso}^{(\alpha)}$. In the numerical simulations in (B)-(D), the parameters were set to $n = 20, \rho = 2, S_{env} = 0.03, V_{env} = 3.0, D_S^{(env)} = 20.0, D_S = 1.0, R_{deg} = 5 \times 10^{-5}, n_{enzyme} = n/5$.

metabolites even under a single-nutrient condition, even though the cell species exhibit different growth rates when grown in isolation (Fig 3C and 3D). Because of this adaptive metabolite exchange, the growth rates of the different species become equal and higher than that exhibited by each species when grown alone (Fig 3D). Thus, symbiosis at the community level is achieved by cell-level adaptation.

Fig 3C and 3D also shows that every cell species leaks some metabolites and consumes others, and metabolites are exchanged between all cells as a result of cell-level adaptation. Unlike the assumptions of food chains or the black queen hypothesis [20, 26, 27], the leaker-consumer relations via metabolic exchanges are entangled, and not hierarchical or cyclic. Hence, no clear trophic levels are observed.

## Cell-level adaptation via leak advantage frequently leads to symbiosis

We then statistically examined how frequently such symbiosis was realized (Fig 4). Note that, in our model, the specific species that ultimately coexist (Fig 3) depend on the order in which new species invaded, as has also been observed for actual microbes [37, 38], whereas the statistics on the symbiosis are reproducible and independent of such ordering.

Fig 4A shows the frequency of symbiotic coexistence of multiple species in a single-nutrient condition. As the number of chemical components $n$ increases, the coexistence of more species

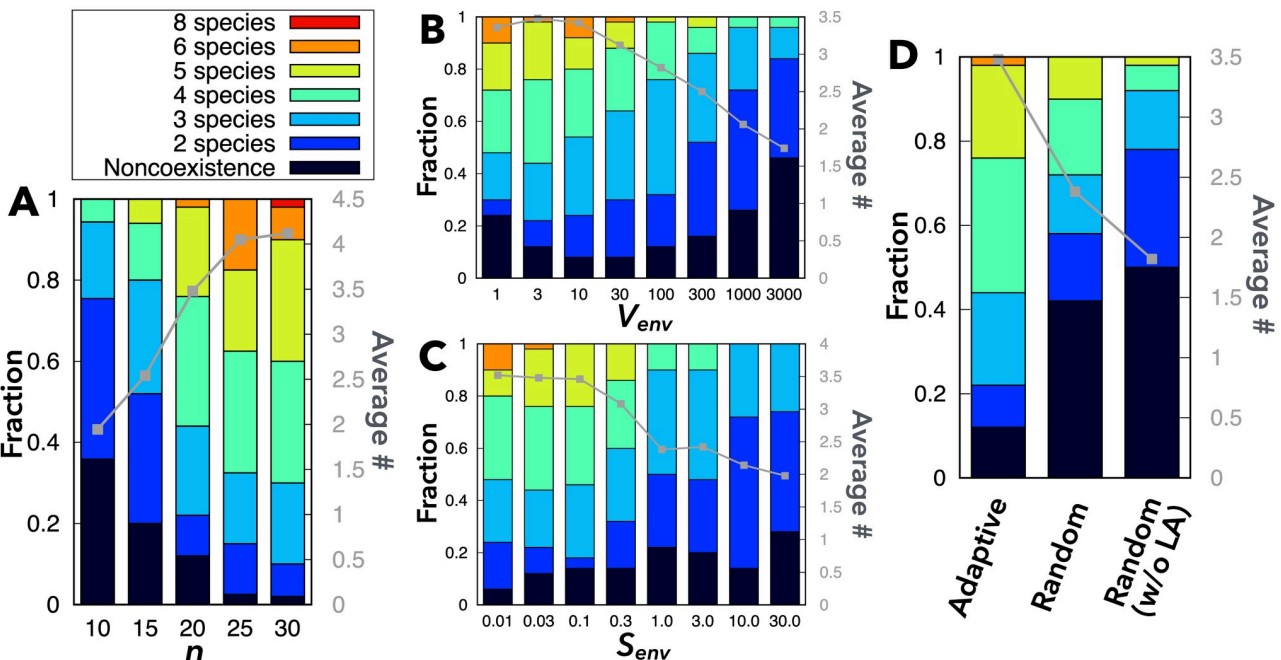

**Fig 4. Statistics of symbiosis among randomly generated networks.** (A) Dependence of the frequency of coexisting species on the number of chemical components $n$. $S_{env} = 0.03$, $V_{env} = 3$. (B) Dependence of the frequency of coexisting species upon $V_{env}$. $n = 20$, $S_{env} = 0.03$. (C) Dependence of the frequency of coexisting species upon $S_{env}$. $n = 20$, $V_{env} = 3$. (D) The frequency of coexisting species for random fixed diffusion coefficients (with and without leakage of chemicals that confer leak advantage) without cell-level adaptation. $n = 20$, $S_{env} = 0.03$, $V_{env} = 3$. In the "random" case, the diffusion coefficients of chemicals $n_{enzyme} + 1 \sim n - 1$ are chosen randomly from a uniform distribution [0.0: 1.0] (see also S6 and S7 Figs). In all the panels, the colored bars show the frequency of symbiosis among two to eight species (with different colors), whereas the black bars show noncoexistence. The frequency for each parameter set was calculated from 50 independent samples of $N$ catalytic networks where the species with the fastest growth in isolation has a leak-advantage chemical in its reaction network. In all the numerical simulations, the other parameters are fixed: $D_S^{(env)} = 20.0$, $D_S = 1.0$, $R_{deg} = 5 \times 10^{-5}$, $n_{enzyme} = n/5$.

is more likely. For $n = 30$, symbiotic coexistence is achieved for almost all the trials, as long as the cell species with the fastest growth in isolation has at least one leak-advantage chemical. Here, recall that the frequency of leak advantage in isolation increases with $n$ [32].

In addition, symbiosis is achieved frequently with a wide range of environmental parameters $S_{env}$ and $V_{env}$ (and $R_{deg}$) (Fig 4B and 4C, and S4 Fig). Notably, Fig 4B demonstrates that the frequency of symbiosis decreases as the size of the environment $V_{env}$ increases, unless the environment is too small ($V_{env} \simeq 1$, i.e., the total volume of cells equals that of the environment). Note here that increase in $V_{env}$ (and/or $R_{deg}$) weakens cell-cell interactions because the secreted chemicals are diluted; consequently, the growth change due to consumption is suppressed while that due to leakage is little affected. Hence, $1/V_{env}$ (and $1/R_{deg}$) serves as an indicator of the strength of cell-cell interactions (or the efficiency of exchange of secreted metabolites). Indeed, for larger $V_{env}$, symbiosis is achieved less frequently by metabolite exchange via the environment. Note that when $V_{env}$ is too small ($V_{env} \simeq 1$), the environmental concentration of chemicals is sensitive to addition of new species, and therefore, the coexistence of multiple species becomes unstable.

Fig 4C shows that decrease in the nutrient supply $S_{env}$ results in increase in the frequency of symbiosis and the average number of coexisting species. This may be explained by the increasing importance of the metabolic efficiency of converting the nutrient into biomass on the growth rate with large $S_{env}$, relative to the efficiency of the exchange of non-nutrient metabolites. Hence, a cell species with slower growth in isolation (which is based on its metabolic

efficiency for the nutrient) may not be able to achieve the same growth rate as those with faster growth in isolation.

In short, the results shown in Fig 4A–4C suggest that symbiosis among multiple species via adaptive metabolite exchange is commonly achievable because microbes contain many chemical components and often exist in crowded and nutrient-poor environments.

Although we have considered the situations where cell species can control their leakiness of each chemical independently, the arguments presented above regarding cell-level adaptation through leak advantage are still valid even when the leakage rates of several chemicals are changed simultaneously. Indeed, even if leakage of all metabolites is regulated by only a single diffusion coefficient (i.e., $D_i^{(\alpha)}$'s for all metabolite $i$ are identical), symbiosis due to leaker-consumer mutualism can be still achieved (S5 Fig).

## Decrease in species diversity in the absence of cell-level adaptation

Furthermore, to ascertain the contribution of cell-level adaptation to symbiosis (via leak advantage), we also considered situations where the leakage and uptake of metabolites occur simply because of the inevitable permeability of cellular membranes, and compared them with the results obtained under the assumption of cell-level adaptation. In this case, the diffusion coefficients of invading cells are given fixed, random positive values for all metabolites (S6 Fig). Interestingly, the frequency of coexistence of multiple species is much smaller (Fig 4D).

Notably, when adaptive changes in the diffusion coefficients at the cell level are allowed, leakage occurs only when it promotes the growth of the leaker cell species (otherwise, the leaker cells would decrease their diffusion coefficient to zero). A mutualistic relationship is thus necessarily established between leaker and consumer cells. Mutualistic cell-cell interaction usually leads to stable coexistence, as discussed in previous studies [23, 39, 40]. In contrast, leakage of the metabolites by random fixed diffusion coefficients does not necessarily benefit leaker cells, thereby making leaker-consumer interactions often parasitic or even competitive (see S7(B) Fig). Such parasitic or competitive interactions often make the steady state (linearly) unstable [41, 42]. Consistently, for the case with randomly pre-fixed diffusion coefficients, the frequency of parasitic (symbiotic) relationships decreases (increases) as the number of coexisting species increases (S7(B) Fig). Hence, the likelihood of coexistence is lower when the diffusion coefficients are fixed.

To further corroborate these results, we also generated a model with null diffusion coefficients for chemicals that confer leak advantage (in isolated conditions) and positive random coefficients for chemicals that do not. In this situation, leakage is always disadvantageous in isolated conditions. In the absence of leakage of leak-advantage chemicals, the frequency of symbiosis and the average number of coexisting species were further reduced (Fig 4D), even though growth promotion due to the uptake of metabolites could still occur.

## Resilience of symbiosis mediated by metabolite exchange

Thus far, we investigated if and how symbiosis of diverse cell species with tangled forms of metabolite exchange is achieved as a result of cell-level adaptation of diffusion coefficients. Lastly, we examined the stability of communities consisting of diverse cell species that exchange metabolites.

In particular, we focused on the resilience of symbiotic relationships against the removal of a coexisting species in the community. In most cases, the removal of one species from the community does not cause the successive extinction of any other species in the community (Fig 5A). In contrast to the extinction of species in a hierarchical ecosystem with trophic levels, where the removal of some *keystone/core species* leads to an *avalanche* of extinctions of

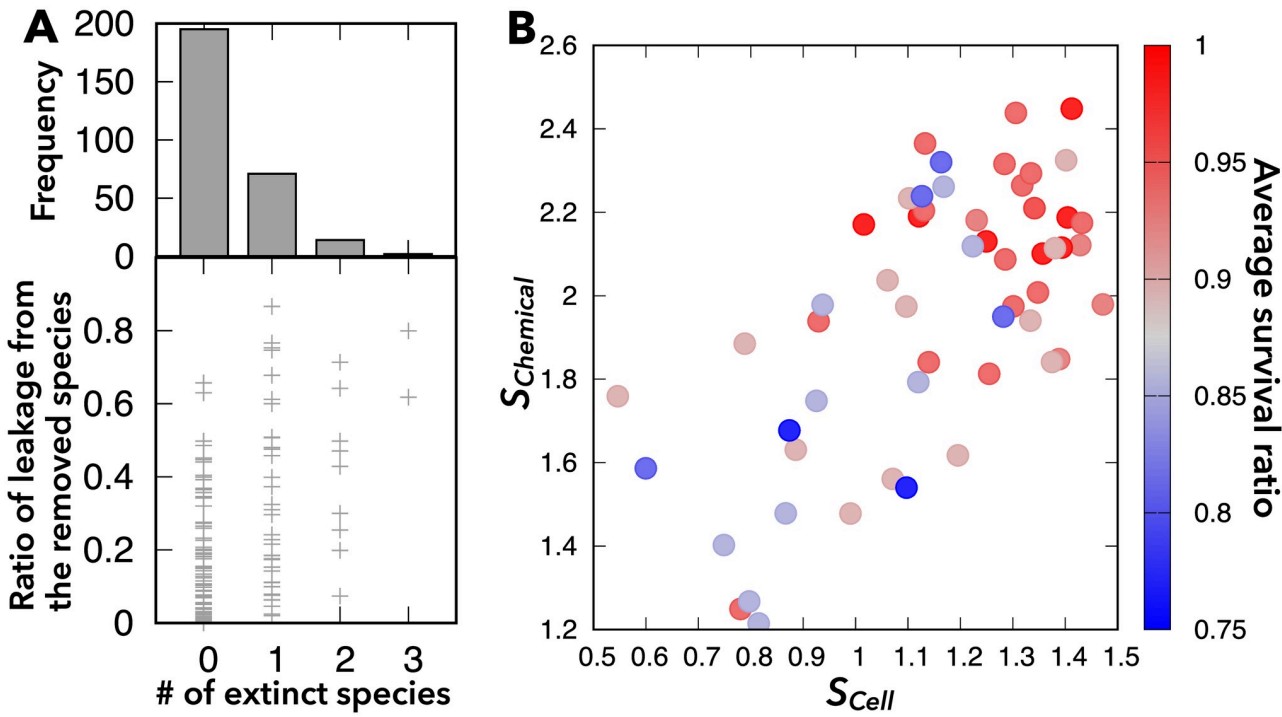

**Fig 5. Resilience of symbiotic coexistence against the removal of one species.** (A) The upper panel shows the frequency distribution of the number of cell species (0–3) that become extinct when one species is removed. The lower panel shows the ratio of leakage of chemicals from the removed cell species to the total leakage from all cell species to the environment, $P_{\text{Leak},\alpha}$. Each point corresponds to a sample shown in the upper panel. (B) Average survival ratio (color) against the indices that characterize the diversity of leaking cell species ($S_{\text{Cell}}$) and leaked chemical components ($S_{\text{Chem}}$). The survival ratio is the number of surviving species after removal of one species, divided by the number of species before the extinction successive to the removal. (If no additional cell species become extinct, this ratio equals one.) The multiple correlation coefficient between survival ratio and ($S_{\text{Cell}}$, $S_{\text{Chem}}$) is 0.49, while the correlation coefficients between the survival ratio and $S_{\text{Cell}}$, between the survival ratio and $S_{\text{Chem}}$, and between $S_{\text{Cell}}$ and $S_{\text{Chem}}$ are 0.46, 0.43, and 0.65, respectively. For these calculations, we used 55 samples of coexistence of five or six species with $n = 20$.

downstream species in the hierarchy [20, 43, 44], such an avalanche of species extinctions hardly occurs in our model (Fig 5A). In other words, we rarely observe the existence of keystone/core species whose absence prevents many other species from coexisting, and removal of which leads to the extinction of all other species. Hence, the present system with metabolite exchange has a high degree of resilience.

As shown in Fig 5A, we did observe a few cases in which removal of one species causes the extinction of most species. In such non-resilient cases, the removed species tended to be ones that dominantly leaked chemicals into the environment. In general, the resilience of the system to removal of species increased with the extent of entanglement of metabolite exchange across cells (i.e., many cell species leak many chemical components) (see also S8 Fig). To quantitatively characterize this tendency, we introduced the indices of the effective "entropies" that characterize the diversity of leaking cell species and leaked chemical components as

$$
S_{\text{Cell}} \quad \equiv \quad -\sum_{\text{Cell }\alpha} P_{\text{Leak},\alpha} \log P_{\text{Leak},\alpha}, \quad S_{\text{Chem}} \equiv -\sum_{\text{Chem. }i} P_{\text{Leaked},i} \log P_{\text{Leaked},i},
$$

where the total flux of all chemicals from cell species $\alpha$ and the total flux of chemical $i$ from all cell species to the environment are defined as $P_{\text{Leak},\alpha} \equiv \sum_i' p_\alpha f_i^{(\alpha)}/L_{\text{tot}}$ and $P_{\text{Leaked},i} \equiv \sum_\alpha' p_\alpha f_i^{(\alpha)}/L_{\text{tot}}$, respectively, with $L_{\text{tot}} \equiv \sum_{(\alpha,i)}' p_\alpha f_i^{(\alpha)}$, where the summation $\Sigma'$ is taken only for leakage (not for uptake), that is $f_i^{(\alpha)} > 0$. If $N$ cell species leak chemicals evenly, $S_{\text{Cell}}$ takes $\log N$ (maximal value),

and if $n$ chemicals are leaked evenly, $S_{Chem}$ takes log $n$. In contrast, if only one cell species leaks or only one chemical is leaked, $S_{Cell}$ or $S_{Chem}$ is equal to 0.

Indeed, when a large number of cells leak and exchange a large number of chemical components (i.e., both $S_{Cell}$ and $S_{Chem}$ are large), the system tends to be resilient (Fig 5B). $S_{Cell}$ and $S_{Chem}$ are also strongly correlated (Fig 5B), such that when many cell species contribute to leakage, many components are leaked, and vice versa. We now consider two extreme situations to illustrate this correlation: if $S_{Chem} = 0$ (i.e., only one chemical is leaked to the environment), then $S_{Cell}$ must be zero because only a single species can exist when only a single niche is available (Gause's competitive exclusion principle) [45–47]. In contrast, if $S_{Chem}$ is large (i.e., many chemicals are leaked), $S_{Cell}$ is unlikely to be zero because a large $S_{Chem}$ allows the coexistence of many cells that can leak some chemicals in turn and thereby increases $S_{Cell}$.

## Discussion

In this paper, we presented a new hypothesis describing how symbiosis mediated by the exchange of various metabolites among diverse cell species is possible, based on the advantages of metabolite leakage for leaker cells. Resilient symbiosis among diverse species can be achieved when each cell species can adaptively change the degree of leakage and uptake of metabolites for its own growth.

First, we described the mechanism and conditions for mutualism between leaker and consumer cells. As the density of leaker cells (with leak advantage) is increased, the leaked metabolites accumulate in the environment, thereby preventing further leakage. Consequently, if a different species that consumes the leaked metabolites for its own growth coexists, such consumption also brings about further leak advantage for the leaker cells. In this leaker-consumer mutualism, both cell species increase their growth rates through cell-cell interactions mediated by secreted metabolites. Although a well-mixed environment and passive diffusion were considered in our model, the basic ideas of leak advantage and leaker-consumer mutualism would depend neither on the spatial structure nor on the means of transport. We thus expect that our theory is applicable not only to marine microbial communities [48, 49] and (synthetic) communities in chemostats or test tubes [17, 22, 50] but also to microbial communities in structured environments such as biofilms [51] and soil [5]. The degradation rate $R_{deg}$ may depend on these habitats (and the feasibility of leaker-consumer mutualism depends on such an environment).

In contrast, whether the leakage is beneficial for the leaker cells is not fully addressed in the previous studies on microbial ecology [26–29]. Though the black queen hypothesis also discusses the evolution of metabolite-mediated cooperation [27, 52], it is premised on metabolite leakage by some species and often (implicitly) assumes that the leakage is just an inevitable consequence of a permeable membrane, and thus the leakage is not necessarily advantageous for the leaker species. Although this assumption is not unreasonable and is consistent with some empirical observations [27, 53, 54], it is not clear why the leaker species have not evolved mechanisms to suppress such leakage that may be disadvantageous and exploited by parasites (free riders). In this respect, our results will complement the black queen hypothesis: some microbial cells secrete chemicals just because this process is beneficial for them. In other words, the "richer" cells "donate" their products to "poorer" cells for the sake of the former cells themselves, as if the cells are practicing *potlatch*, a gift-giving ritual in the human society [55, 56]. In this ritual practiced by indigenous peoples of the North American Pacific Northwest, they increase the competitive gift-giving, although the givers in the actual potlach receive the prestige rather than an immediate benefit from the act.

This "microbial potlatch" generally emerges as a result of individual-level adaptation under conditions of complex intracellular metabolic networks and crowded environments. Indeed, facilitation of growth due to the *de novo emergence of coexistence among* different strains or species has been reported in several experiments [7, 21–23], including those conducted under nutrient depletion [24, 57–59]. Remarkably, in our simulations, adverse, resource-limited conditions facilitate symbiosis among multiple species. Although some studies have reported that lower resource availability leads to a more diverse community [60, 61], other studies have reported contradictory results [62, 63]. Note here that the existence of wasteful or inhibitory byproducts, which was not assumed in our simulations, could lead to greater diversity under nutrient-rich environments, as nutrient excess will increase the waste byproducts [64, 65]. Indeed, the actual relationship between diversity and resource availability is sometimes non-monotonous [66, 67] and can depend on the cultivation conditions [60]. This complex relationship may be a consequence of the competing effects of essential metabolites and waste byproducts.

Moreover, symbiotic coexistence is more frequent when the diffusion coefficients are variable due to cell-level adaptation. When diffusion coefficients are fixed, even if all metabolites are leaked into the environment and thus the number of niches is larger, we found that the actual number of coexisting species in our simulations is much lower.

Furthermore, we examined the ecological resilience of microbial communities against the removal of a coexisting species. As more cells leak and exchange more chemical components in an ecosystem (i.e., $S_{Cell}$ and $S_{Chem}$ are larger), the microbial community becomes more resilient to the removal of a member. In contrast, if only a few cell species leak a few chemicals, the ecosystem would have a unidirectional structure similar to a "food chain." In such cases, keystone/core species could exist, and the system would not be resilient to removal of such species. Empirical studies suggest that diverse microbial communities are more resistant to environmental disturbances than monocultures [40]. Experimentally estimating $S_{Cell}$ or $S_{Chem}$ and investigating their relationship with the resilience of microbial communities could be interesting.

Theoretically, we find that coexistence of diverse species is achieved by incorporating multi-level dynamics at the intercellular (population) and intracellular (metabolic) levels that cannot be captured by standard Lotka–Volterra-type population dynamics. Microbial ecosystems with metabolite exchange via the environment are expected to behave differently from those with simple food chain or food web structures that are often considered in Lotka–Volterra-type population dynamics, as the interactions between different cell species depend not only on their populations but also on the exchanged chemicals, which depend on their intracellular states [68]. For example, in leaker-consumer mutualism, the benefit for leaker cells is indirect; the leaker cells benefit from the consumption of accumulated chemicals only when the density of leaker cells is high enough to cause excess accumulation of these chemicals. The leaker-consumer mutualism is thus frequency-dependent and depends on the degree of interaction between cells via secreted chemicals. In the present model, this degree of cell-cell interaction depends on the relative volume of the medium to that of a cell $V_{env}$ (i.e., the inverse of cell density) and the degradation rate of chemicals in the medium $R_{deg}$. If $V_{env}$ and $R_{deg}$ are sufficiently large, the leaker cells can continue to leak chemicals and grow efficiently without consumer cells. In this sense, leaker-consumer mutualism with unidirectional flow is different from ordinary forms of metabolic division of labor [35, 69]. Still, in a system with multiple cell species we studied here, each cell can simultaneously be a leaker and a consumer for different chemicals, thus achieving metabolic division of labor with bidirectional exchange. These explain how coexistence and symbiosis of diverse microbes with chemical exchange can be spontaneously realized and maintained through adaptation and evolution, whereas the existing

coexistence mechanisms that are based on trade-offs between different ecological functions, such as differential use of resource types [70, 71], or on nonlinear predator-prey dynamics [72] cannot explain how such mutualism emerges and is maintained through chemical exchange.

The premise of the present study was that the leakage of even essential metabolites can be beneficial for cellular growth under certain conditions, and the control of leakage provides a possible means of adaptation. Indeed, there are many chemicals that are nutritional at low concentrations but can be inhibitory when abundant [9]; such examples might be understood as a consequence of leak advantage, and at least, can lead to leak-consumer mutualism in the same way. In principle, the theory of leak advantage could be testable by confirming the increases (decreases) in the cellular growth rate against the leakage flux (the extracellular concentration) through experiments on microbes in which the concentration of some secreted chemical (e.g., amino acids or organic acids) in the culture medium is controlled by means of a chemostat. Moreover, a recent phylogenic study revealed that leakage of essential metabolites has been acquired and promoted through evolution [30]; also, promotion of metabolite leakage and exchange is observed in laboratory evolution [31]. These suggest that the regulation of leakiness is actually utilized in adaptation and evolution [22]. At least, a minimal model of actual metabolic systems adopted from [73], rather than random networks, can show leak advantages of essential metabolites (S1 Fig), and the values of some parameters used in the present study such as $V_{env} \lesssim 1000$, $\rho \lesssim 2$ seem realistic [74]. However, confirmation of leak advantages and resultant symbioses by simulations of detailed reaction networks of real microbes as well as experiments will be important problems to be explored.

The coexistence of multiple species via active secretion of chemicals is considered in the context of classical syntrophy in microbial communities [15, 75], where it is generally assumed that the leaked chemicals are wasteful or inhibitory to the leaker species itself but are useful for another specific species. Such chemicals could surely exist [18] but the leaked essential chemicals would be likely to be useful for more diverse (non-specific) species (which would allow for the coexistence of more diverse species). Moreover, the leaked chemicals in classical syntrophy are thought to be located at a lower level of the chemical hierarchy according to energetics [8, 15], which would also determine the trophic hierarchy. In contrast, chemicals that confer leak advantage are often essential, and lead to entangled networks of metabolite exchanges between different cell species, as often observed in actual microbial ecosystems [5, 10, 17]. However, it is of note that coexistence among diverse species will become more frequent when the inhibitory effect of waste byproducts is additionally included in our model.

Finally, let us discuss whether a leak advantage can be maintained through the course of evolution that may incorporate appropriate gene-regulation of enzymatic activity, which is a well-known means to optimize cell growth [76]. If only a single cell species exists, such optimization of gene regulation would be possible to eliminate the leak advantage by such evolution. However, if cells interact with other cells, it would be difficult to reach such an optimized growth state without leakiness, through evolutionary optimization in enzymatic activity. As the cell numbers increase, the environment inevitably becomes crowded, and cell-cell interactions through secreted chemicals will not be easily eliminated. Hence, optimization under isolated conditions would not progress unless cells find an optimized solution without any secretion of chemicals. Once the evolution progresses under the environmental conditions with interacting cells, finding such a solution, even if it exists, would take many generations. Before such isolated optimization is reached, other cells that consume secreted chemicals could either emerge through mutation or invade from elsewhere, thereby enhancing the growth of the leaker species. Then, symbiotic relationships with different cell species will develop and result in further entanglement of chemical exchange networks, as described in this paper.

In summary, we have shown that cell-level adaptation of leakiness of (essential) metabolites spontaneously establishes symbiotic relationships. This "microbial potlatch" generally emerges when the intracellular metabolic network is complex, the environment is crowded, and nutrient supply is limited. The present study thus provides a basis for complex microbial ecosystems with diverse species.

## Supporting information

**S1 Fig. Leak advantage with a "realistic" metabolic reaction network.** (A) The network structure of a minimal bioreaction model adopted from [73]. Each rounded box and filled square represents a metabolite and a reaction, respectively, and the numbers along some arrows indicate stoichiometric coefficients. For simplicity, all the rate constants are set at unity, and biomass is assumed to be made from nucleotides and alanine; accordingly, the growth rate is given by $\mu(\mathbf{x}) \equiv x_{\text{Nucleotides}} x_{\text{Alanine}}$. Glucose and glutamine are externally supplied as nutrients: their diffusion coefficients $D_{\text{Glucose}}$, $D_{\text{Glutamine}}$ and external concentrations $x_{\text{Glucose}}^{(\text{env})}$, $x_{\text{Glutamine}}^{(\text{env})}$ are fixed at 1.0, while the rest of the external concentrations $\mathbf{x}^{(\text{env})}$ are fixed at 0.0. The metabolites highlighted by pink are leak-advantage metabolites in the isolated condition, i.e., (moderate) leakage of them promote the cell growth, while the leakage of CO2, NH4, or lactate is neutral in our numerical simulations since they are at the bottom of the network. (B) Dependence of growth rate $\mu$ on the diffusion coefficient of some leak-advantage metabolites: Glucose-6-P (G6P), Pyruvate (Pyr), and Erythrose-4-P (E4P). The horizontal line in each panel exhibits the growth rate with no leakage of non-nutrient metabolites ($\mu = 0.0058$). (PDF)

**S2 Fig. Leak advantage and leaker-consumer mutualism with reversible chemical reactions.** The network in Fig 2 is utilized, while the chemical reactions $S + E \rightarrow M_1 + E$ and $S \rightarrow M_2$ are reversal with the strength $r$, i.e., $k_{M_1 \rightarrow S} = rk_{S \rightarrow M_1}$ and $k_{M_2 \rightarrow S} = rk_{S \rightarrow M_2}$. (A) Dependence of the isolated leaker's growth rate on $D_{M_1}$ with the the reversibility $r = 0.0, 0.1, 1.0$. (B) Phase diagram of leak advantage of $M_1$ for the isolated leaker cell species depending on the reversibility $r$ and nutrient supply $S_{\text{env}}$. Regions LA (red) and NoLA (black) are delineated by gray lines and represent whether the leak advantage of $M_1$ exists or not, respectively. (C) Phase diagram of symbiosis depending on $D_{M_1}^{(\text{leaker})}$ and $D_{M_1}^{(\text{consumer})}$ with the reversibility $r = 0.01$. Regions M (red) and NC (black) are delineated by gray lines and represent mutualism and noncoexistence, respectively. In the numerical simulations in (A)-(C), the rate constants are set as: $k_{S \rightarrow M_1}^{(\text{leaker})} = 1, k_{S \rightarrow M_1}^{(\text{consumer})} = 0.3, k_{M_1 \rightarrow \text{rb}}^{(\text{leaker})} = k_{M_1 \rightarrow E}^{(\text{leaker})} = k_{S \rightarrow M_2}^{(\text{leaker})} = 1, k_{M_1 \rightarrow \text{rb}}^{(\text{consumer})} = k_{M_1 \rightarrow E}^{(\text{consumer})} = k_{S \rightarrow M_2}^{(\text{consumer})} = 2,$ $k_{M_2 \rightarrow \text{BM}}^{(\text{leaker})} = k_{M_2 \rightarrow \text{BM}}^{(\text{consumer})} = 0.01$. The other parameters are set as $S_{\text{env}} = 1.0, D_S^{(\text{env})} = 10.0, D_S = 1.0,$ $V_{\text{env}} = 1.0, R_{\text{deg}} = 0.1$. (PDF)

**S3 Fig. Examples of randomly generated networks.** (A) Example of two-species symbiosis among randomly generated networks with $n = 10$, $\rho = 2$ and metabolite exchange between them. (B) Example of randomly generated networks with $n = 20$, $\rho = 2$ (cell species B in Fig 3C and 3D). The enzyme labeled on each arrow catalyzes the conversion of the metabolite at the arrowtail to the metabolite or enzyme at the arrowhead. Among $n$ chemicals, chemicals 1 and $n_{\text{enzyme}} = n/5$ are enzymes (green squares) and the nutrient chemical 0 and chemicals 3 to $n - 1$ are metabolites (orange circles). The leak-advantage metabolites for each cell species in isolation are highlighted by pink and adaptively leaked into the environment. (PDF)

**S4 Fig. Dependence of the frequency of coexisting species upon $R_{deg}$.** The colored bars show the frequency of symbiosis among two to six species (shown in different colors), whereas the black bars show noncoexistence. The frequency for each parameter set was calculated from 50 independent samples of $N$ catalytic networks where the species with the fastest growth in isolation has a leak-advantage chemical in its reaction network. The other parameters are fixed: $n = 20, S_{env} = 0.03, V_{env} = 3.0, D_S^{(env)} = 20.0, D_S = 1.0, n_{enzyme} = n/5$.
(PDF)

**S5 Fig. Dependence of the frequency of coexisting species against the changes in the specificity of metabolite leakage (i.e., the degrees of freedom for diffusion coefficients in each cell species).** The value for "specificity" is defined as the degrees of freedom in the adaptation of the diffusion coefficients; for example, the case with specificity = 15 corresponds to the case in which each cell can alter the diffusion coefficients for all 15 non-nutrient metabolites independently; while in the case with specificity = 1, each cell species alters the diffusion coefficients of each metabolite all together across all metabolites, i.e., the diffusion coefficients of all metabolites are identical. When specificity equals an intermediate value 3 (5), each cell species has 3 (5) different values for the diffusion coefficients and alters the diffusion coefficients for 15/specificity = 5 (15/specificity = 3) non-nutrient metabolites together. The colored bars show the frequency of symbiosis with the number of coexisting species for two to six species (presented in different colors), whereas the black bars show noncoexistence. The frequency for each parameter set was calculated from 50 independent samples of $N$ catalytic networks where the species with the fastest growth in isolation has a leak-advantage chemical in its reaction network. In the numerical simulation, the parameters were set to $n = 20, S_{env} = 0.03, V_{env} = 3.0, D_S^{(env)} = 20.0, D_S = 1.0, R_{deg} = 5 \times 10^{-5}, n_{enzyme} = n/5$.
(PDF)

**S6 Fig. Example of coexistence with randomly pre-fixed diffusion coefficients.** The diffusion coefficients of all the non-nutrient metabolites (chemicals $n_{enzyme} + 1$ to $n - 1$) were randomly chosen from a uniform distribution [0.0: 1.0]. (A) Time series of the number of coexisting species by successful invasions of new species and the growth rate of cell species in coexistence. (B) Plot of leakage (blue) and uptake (red) fluxes of non-nutrient chemicals from each cell species A-E. (C) Structure of metabolic exchange among five species that originally have different growth rates in isolation. Cyan and pink arrows indicate the leakage and uptake of each chemical component, respectively. The growth rate while coexisting, $\mu_{symbiosis}$, is indicated on the top blue line. In the numerical simulation, the parameters were set to $n = 20, S_{env} = 0.03, V_{env} = 3.0, D_S^{(env)} = 20.0, D_S = 1.0, R_{deg} = 5 \times 10^{-5}, n_{enzyme} = n/5$.
(PDF)

**S7 Fig. Statistics on coexistence with randomly pre-fixed diffusion coefficients.** (A) Dependence of the frequency of coexisting species on the upper bound of randomly pre-fixed diffusion coefficients. The diffusion coefficients of non-nutrient metabolites (chemicals $n_{enzyme} + 1$ to $n - 1$) were randomly chosen from a uniform distribution [0.0: $D_{max}$]. The frequency for each parameter set was calculated from 50 independent samples of $N$ randomly generated networks where the species with the fastest growth in isolation has a leak-advantage metabolite. (B) Relationship between the frequency of symbiosis/parasitism and the number of coexisting species (with various values of $D_{max}$). In all the numerical simulations, the parameters are set at $n = 20, S_{env} = 0.03, V_{env} = 3.0, D_S^{(env)} = 20.0, D_S = 1.0, R_{deg} = 5 \times 10^{-5}, n_{enzyme} = n/5$.
(PDF)

**S8 Fig. Examples of (A) resilient and (B) non-resilient coexistence.** (A) The removal of any species does not cause the extinction of another species (average survival ratio is 1). (B) The removal of cell species B leads to the extinction of the other three cell species (C,D,E) leaving only cell species A in the community (average survival ratio is 0.8). Blue and red indicate the leakage and uptake of each chemical component, respectively. In both simulations, the parameters are set at $n = 20, S_{env} = 0.03, V_{env} = 3.0, D_S^{(env)} = 20.0, D_S = 1.0, R_{deg} = 5 \times 10^{-5}, n_{enzyme} = n/5$. (PDF)

**S1 Text. Supplementary information about details of the model of symbiosis among randomly generated networks.**
(PDF)

# Acknowledgments

The authors would like to thank Chikara Furusawa and Kazufumi Hosoda for their useful comments.

# Author Contributions

**Conceptualization:** Jumpei F. Yamagishi, Nen Saito, Kunihiko Kaneko.

**Data curation:** Jumpei F. Yamagishi.

**Formal analysis:** Jumpei F. Yamagishi.

**Funding acquisition:** Kunihiko Kaneko.

**Investigation:** Jumpei F. Yamagishi, Nen Saito, Kunihiko Kaneko.

**Methodology:** Jumpei F. Yamagishi, Nen Saito, Kunihiko Kaneko.

**Project administration:** Jumpei F. Yamagishi, Nen Saito, Kunihiko Kaneko.

**Software:** Jumpei F. Yamagishi.

**Supervision:** Nen Saito, Kunihiko Kaneko.

**Visualization:** Jumpei F. Yamagishi, Nen Saito, Kunihiko Kaneko.

**Writing – original draft:** Jumpei F. Yamagishi, Nen Saito, Kunihiko Kaneko.

**Writing – review & editing:** Jumpei F. Yamagishi, Nen Saito, Kunihiko Kaneko.

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
