## [Decision Letter · Decision Letter 0]

27 Apr 2021

Dear Prof. Kaneko,

Thank you very much for submitting your manuscript "Adaptation of metabolite leakiness leads to symbiotic chemical exchange and to a resilient microbial ecosystem" for consideration at PLOS Computational Biology. As with all papers reviewed by the journal, your manuscript was reviewed by members of the editorial board and by several independent reviewers. The reviewers appreciated the attention to an important topic. Based on the reviews, we are likely to accept this manuscript for publication, providing that you modify the manuscript according to the review recommendations. (Please note that the comments of Reviewer #3 are attached separately.)

The PLOS Data policy requires that all data and code be made available at the time of publication. Please ensure that your code is available to the scientific community before submitting a revised manuscript.

Sincerely,

Paul Jensen

Guest Editor

PLOS Computational Biology

Alice McHardy

Deputy Editor

PLOS Computational Biology

[LINK]

Reviewer's Responses to Questions

**Comments to the Authors:**

Reviewer #1: In their paper, Yamagishi, Saito, and Kaneko argue that metabolite secretion in mixed communities can lead to mutualism between “leaker” and “consumer” cells in a microbial community. This work contributes to a topic of considerable contemporary interest: What allows complex microbial communities to assemble, and what mathematical frameworks can be used to understand this process? The manuscript is quite well written and easy to follow. I have a small number of questions and comments:

-Line 107: “of the above equation”: Please include explicit equation numbers for clarity.

-Line 155: “randomly generated networks”: I understand that this is a model that is easy to implement computationally, however, can you provide some insight into whether this is an appropriate model for understanding shared metabolic networks in natural systems? For example, have random networks like this been used to understand shared metabolites in the microbiome, etc.?

-Line 169: “After the addition of the new species”: It’s not clear to me that the model will yield the same result if the order of species introduction is altered. Can you comment on the importance of historical contingency (the order in which species are introduced) in your model?

-Line 184: “BQH”: This acronym doesn’t get used all that often. Writing out Black Queen Hypothesis would make the paper more readable.

-Figure 3A: The random network is so large that’s is hard to interpret as a viewer. Perhaps a smaller network (like N = 6, as in panel D) would be more appropriate?

-Line 231: “increases(S1)” Missing space.

-Line 242: “species that coexisted”: Missing an additional clause, such as: “species that coexisted with other species in the community”

-Line 296: “Microbial potlach”: I really like this phrase, but I wonder if more clarification is needed. My, very naïve understanding (I’m not an anthropologist!), is that potlaches were all about showing off your prestige, and not getting an immediate benefit from the act. This seems to be different from your model, so a little more description would be useful before coining a term that may be confusing to many readers.

-Line 346: “is testable”: I agree that testing this model in a laboratory setting would be exciting. However, I wonder if in the interim you could provide some insight into whether the parameters used in your model are within a physiologically relevant range? I know this can be challenging, but such data, however modest, would broaden the relevance of these findings, especially given that your model has a large number of parameters.

-SI material: The high-level description of the code is quite useful, however can you also provide details of how this was implemented, such as software and libraries used for solving these models?

-Code availability: Please provide a link to the code (e.g., a github page) used in your simulations.

Reviewer #2: The authors study a mutualistic system by numerical simulations, where the leakers themselves benefit with enhanced growth rate. The rationale behind such behavior was presented in their previous publication (ref. 32) on this subject. Here they study the behavior of such a system, its domain of emergence and stability. Under the assumption of existence of benefit for the leakers, they show that a complex system of leakers and consumers may be stable and diverse.

The paper is relevant for a broader scientific community, since it advances our understanding of the mechanisms of symbiotic relations in the context of the Black Queen Hypothesis theory. I would like to recommend the paper for publication, given that the authors address my two comments.

1. In the Discussion section they write “Although a well-mixed environment and passive diffusion were considered in our model, the ideas of leak advantage and leaker-consumer mutualism do not depend on the spatial structure and the means of transport. Our theory is thus applicable not only to marine microbial communities (46, 47) and (synthetic) communities in chemostats or test tubes (17, 22, 48) but also to microbial communities in structured environments such as biofilms (49) and soil (5).” I am not convinced that the authors have provided evidence in this work that “the ideas of leak advantage and leaker-consumer mutualism do not depend on the spatial structure and the means of transport.” This may not be true, and I would advise that the authors change this statement in a way that would convey that this is their expectation, for which they have not shown evidence.

2. In the second to last paragraph of the Discussion section starting with “Finally, let us discuss whether a leak advantage would be eliminated through the course of evolution…” they claim that “it would be difficult to reach such an optimized growth state without leakiness, through evolutionary optimization in enzymatic activity.” Again, as in point 1, I am not convinced that this necessarily follows from this work, nor that the authors have shown any evidence for this conclusion. They say that growth optimization that would eliminate leakage is impossible because “Before such isolated optimization is reached, other cells that consume secreted chemicals could either emerge through mutation or invade from elsewhere, there enhancing the growth of the leaker species.” These processes certainly may happen, but not necessarily so, depending on the rates of mutations and/or invasions. It feels to me that this specific question deserves a whole study on its own, and would advise the authors to treat it accordingly.

Reviewer #3: Attached

**Have the authors made all data and (if applicable) computational code underlying the findings in their manuscript fully available?**

Reviewer #1: **No: **I couldn't find a link to the code used in their models.

Reviewer #2: Yes

Reviewer #3: **No: **I could not find the data or the code with the paper

PLOS authors have the option to publish the peer review history of their article (what does this mean?). If published, this will include your full peer review and any attached files.

Reviewer #1: No

Reviewer #2: No

Reviewer #3: **Yes: **Purushottam Dixit

Figure Files:

Data Requirements:

Reproducibility:

References:

---

## [Editor Report · Decision Letter 1]

3 Jun 2021

Dear Prof. Kaneko,

We are pleased to inform you that your manuscript 'Adaptation of metabolite leakiness leads to symbiotic chemical exchange and to a resilient microbial ecosystem' has been provisionally accepted for publication in PLOS Computational Biology.

Best regards,

Paul Jensen

Guest Editor

PLOS Computational Biology

Alice McHardy

Deputy Editor

PLOS Computational Biology

---

## [Editor Report · Acceptance letter]

17 Jun 2021

PCOMPBIOL-D-21-00478R1 

Adaptation of metabolite leakiness leads to symbiotic chemical exchange and to a resilient microbial ecosystem

Dear Dr Kaneko,

I am pleased to inform you that your manuscript has been formally accepted for publication in PLOS Computational Biology. Your manuscript is now with our production department and you will be notified of the publication date in due course.

With kind regards,

Katalin Szabo
